# Measuring and Harnessing Transference in Multi-Task Learning

## Abstract

Multi-task learning can leverage information learned by one task to benefit the training of other tasks. Despite this capacity, naïve formulations often degrade performance and in particular, identifying the tasks that would benefit from co-training remains a challenging design question. In this paper, we analyze the dynamics of information transfer, or *transference*, across tasks throughout training. Specifically, we develop a similarity measure that can quantify transference among tasks and use this quantity to both better understand the optimization dynamics of multi-task learning as well as improve overall learning performance. In the latter case, we propose two methods to leverage our transference metric. The first operates at a macro-level by selecting which tasks should train together while the second functions at a micro-level by determining how to combine task gradients at each training step. We find these methods can lead to significant improvement over prior work on three supervised multi-task learning benchmarks and one multi-task reinforcement learning paradigm.

## 1 Introduction

Deciding if two or more objectives should be trained together in a multi-task model, as well as choosing how that model's parameters should be shared, is an inherently complex issue often left to human experts (Zhang & Yang, 2017). However, a human's understanding of similarity is motivated by their intuition and experience rather than a prescient knowledge of the underlying structures learned by a neural network. To further complicate matters, the benefit or detriment induced from co-training relies on many non-trivial decisions including, but not limited to, dataset characteristics, model architecture, hyperparameters, capacity, and convergence (Wu et al., 2020; Vandenhende et al., 2019; Standley et al., 2019; Sun et al., 2019). As a result, a quantifiable measure which conveys the effect of information transfer in a neural network would be valuable to practitioners and researchers alike to better construct or understand multi-task learning paradigms (Baxter, 2000; Ben-David & Schuller, 2003).

The training dynamics specific to multitask neural networks, namely cross-task interactions at the shared parameters (Zhao et al., 2018), are difficult to predict and only fully manifest at the completion of training. Given the cost, both with regards to time and resources, of fully training a deep neural network, an exhaustive search over the $2^m - 1$ possible combinations of $m$ tasks to determine ideal task groupings can be infeasible. This search is only complicated by the irreproducibility inherent in traversing a loss landscape with high curvature; an effect which appears especially pronounced in multi-task learning paradigms (Yu et al., 2020; Standley et al., 2019).

In this paper, we aim to take a step towards quantifying *transference*, or the dynamics of information transfer, and understanding its effect on multi-task training efficiency. As both the input data and state of model convergence are fundamental to transference (Wu et al., 2020), we develop a parameter-free approach to measure this effect at a per-minibatch level of granularity. Moreover, our quantity makes no assumptions regarding model architecture, and is applicable to any paradigm in which shared parameters are updated with respect to multiple task losses.

By analyzing multi-task training dynamics through the lens of transference, we present the following observations. First, information transfer is highly dependent on model convergence and varies significantly throughout training. Second, and perhaps surprisingly, excluding certain task gradients from the multi-task gradient update for select minibatches can improve learning efficiency. Our

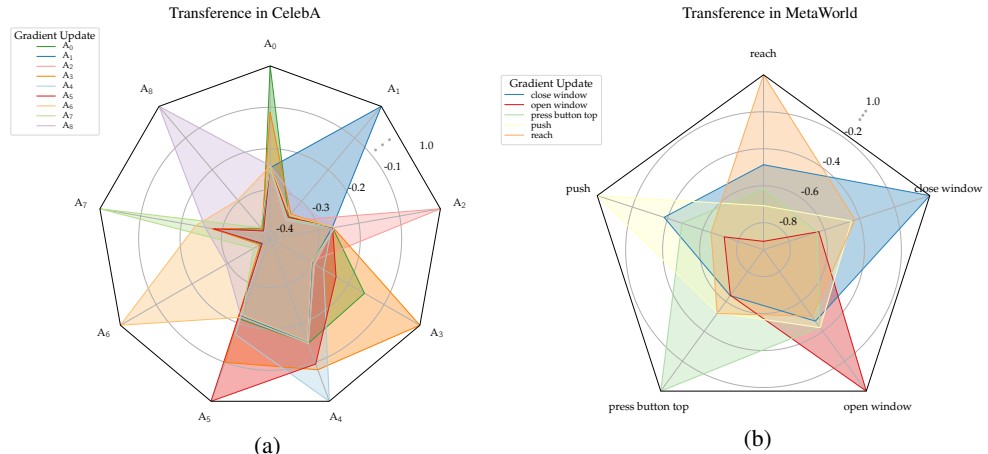

Figure 1: Transference in (a) CelebA for a subset of 9 attributes; (b) Meta-World for "push", "reach", "press button top", and "open window". To determine task groupings, we compute the transference of each task $i$ on all other tasks $j$, i.e. $\mathcal{Z}^t_{\{i\}\to j}$ and average over time. For the purpose of illustration, we normalize the transference along each axis. Notice the majority of the tasks in (a) concentrate around a single value for each attribute. Tasks which exhibit transference above this value are considered to have relatively high transference. For instance, $A_3$ exhibits higher-than-average transference on $A_0$, $A_4$, and $A_5$. A similar effect is observed in (b), with "close window" manifesting high transference on "push" and "reach".

analysis suggests this is due to large variation in loss landscapes for different tasks as illustrated in Figure 4. Building on these observations, we propose two methods to utilize transference in multi-task learning algorithms – to choose which tasks to train together as well as determining which gradients to apply at each minibatch. Our experiments indicate the former can identify promising task groupings, while the latter can improve learning performance over prior methods.

In summary, our main contributions are three-fold: we (1) propose the first measure (to our knowledge) which quantifies information transfer among tasks in multi-task learning; (2) demonstrate how transference can be used as a heuristic to select task groupings; (3) present a method which leverages minibatch-level transference to augment network performance.

## 2 RELATED WORK

**Multi-Task Formulation.** The most prevalent formulation of MTL is *hard parameter sharing* of hidden layers (Ruder, 2017; Caruana, 1993). In this design, a subset of the hidden layers are typically shared among all tasks, and task-specific layers are stacked on top of the shared base to output a prediction value. Each task is assigned a weight, and the loss of the entire model is a linear combination of each task's loss multiplied by its respective loss weight. This particular design enables parameter efficiency by sharing hidden layers across tasks, reduces overfitting, and can facilitate transfer learning effects among tasks (Ruder, 2017; Baxter, 2000; Zhang & Yang, 2017).

**Information Sharing.** Prevailing wisdom suggests tasks which are similar or share a similar underlying structure may benefit from co-training in a multi-task system (Caruana, 1993; 1997). A plethora of multi-task methods addressing *what to share* have been developed, such as Neural Architecture Search (Guo et al., 2020; Sun et al., 2019; Vandenhende et al., 2019; Rusu et al., 2016; Huang et al., 2018; Lu et al., 2017) and Soft-Parameter Sharing (Misra et al., 2016; Duong et al., 2015; Yang & Hospedales, 2016), to improve multi-task performance. Though our measure of transference is complementary with these methods, we direct our focus towards which tasks should be trained together rather than architecture modifications to maximize the benefits of co-training.

While deciding which tasks to train together has traditionally been addressed with costly cross-validation techniques or high variance human intuition, recent advances have developed increasingly efficient algorithms to assess co-training performance. Swirszcz & Lozano (2012) and Bingel & Søgaard (2017) approximate multi-task performance by analyzing single-task learning characteristics. An altogether different approach may leverage recent advances in transfer learning focused on understanding task relationships (Zamir et al., 2018; Achille et al., 2019b; Dwivedi & Roig, 2019; Zhuang et al., 2020; Achille et al., 2019a); however, Standley et al. (2019) show transfer learning

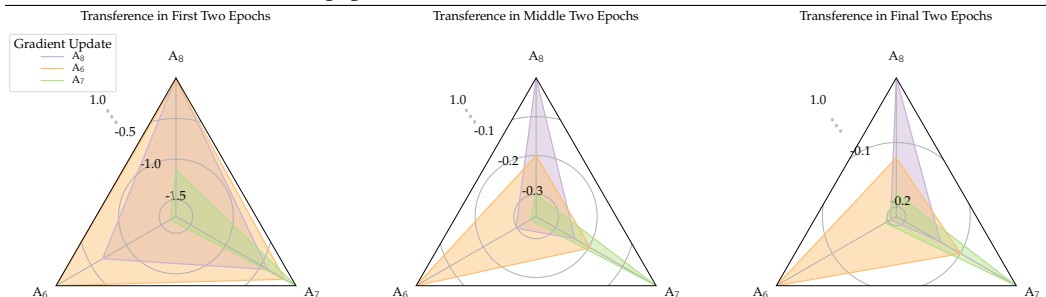

Figure 2: Effect of convergence on transference for CelebA attributes $A_6$, $A_7$, and $A_8$. Transference is highly dynamic and changes over the course of training. See Section 5.2 for more details.

algorithms which determine task similarity do not carry over to the multi-task learning domain and instead propose a multi-task specific framework.

**Training Dynamics.** Significant effort has also been invested to improve the training dynamics of MTL systems. In particular, dynamic loss reweighing has achieved performance superior to using fixed loss weights found with extensive hyperparameter search (Kendall et al., 2018; Guo et al., 2018; Liu et al., 2019; Chen et al., 2017; Sener & Koltun, 2018; Lin et al., 2019). Another set of methods seeks to mitigate the optimization challenges in multi-task learning by manipulating the task gradients in a number of ways such as (1) modifying the direction of task gradients with the underlying assumption that directional inconsistency of gradients on the shared parameters are detrimental to model convergence and performance (Zhao et al., 2018; Suteu & Guo, 2019), and (2) altering both the direction and the magnitude of the task gradients (Yu et al., 2020; Chen et al., 2020; Wang et al., 2020b). Instead of directly modifying the task gradients during optimization, our work builds upon these approaches by quantifying how a gradient update to the shared parameters would affect training loss and choosing the combination of gradients which maximizes positive information transfer.

**Looking into the Future.** Looking at *what could happen* to determine *what should happen* has been used extensively in both the meta-learning (Finn et al., 2017; Nichol et al., 2018; Brinkmeyer et al., 2019; Grant et al., 2018; Kim et al., 2018) as well as optimization domains (Nesterov, 1983; Hinton & Plaut, 1987; Zhang et al., 2019; Izmailov et al., 2018; Johnson & Zhang, 2013). *Lookahead* meta-learning algorithms focusing on validation loss have also been used to improve generalization in multi-task learning systems (Wang et al., 2020a), and our work further adapts this central concept to multi-task learning to both quantify and improve information transfer.

## 3  TRANSFERENCE IN MULTI-TASK LEARNING

Within the context of a hard-parameter sharing paradigm, tasks collaborate to build a shared feature representation which is then specialized by individual task-specific heads to output a prediction. Accordingly, tasks implicitly transfer information to each other by updating this shared feature representation with successive gradient updates. We can then view transference, or information transfer in multi-task learning, as the effect of a task's gradient update to the shared parameters on another task's loss during training.

While the the shared parameter update using a task's gradient, often but not always, increases the losses of the other tasks in the network, we find the extent to which these losses change to be highly task specific. This indicates certain tasks interact more constructively than others. Further, we notice this effect to be reproducible and nearly unchanged across successive training runs with varying parameter initializations. Motivated by these observations, we derive a quantitative measure of transference, describe how it can be used to determine which tasks should be trained together, and provide empirical analysis of these claims. Later, we will build upon these ideas to derive a new multi-task learning algorithm.

### 3.1  MEASURING TRANSFERENCE

Consider an $m$-multitask loss function parameterized by $\{\theta_s\} \cup \{\theta_i \,|\, i \in [m]\}$ where $\theta_s$ represents the shared parameters and $\theta_i$ represents the task $i \in [m]$ specific parameters. Let

$$L_{\text{total}}(\mathcal{X}, \theta_s, \{\theta_i\}) = \sum_{i \in [m]} L_i(\mathcal{X}, \theta_s, \theta_i),$$

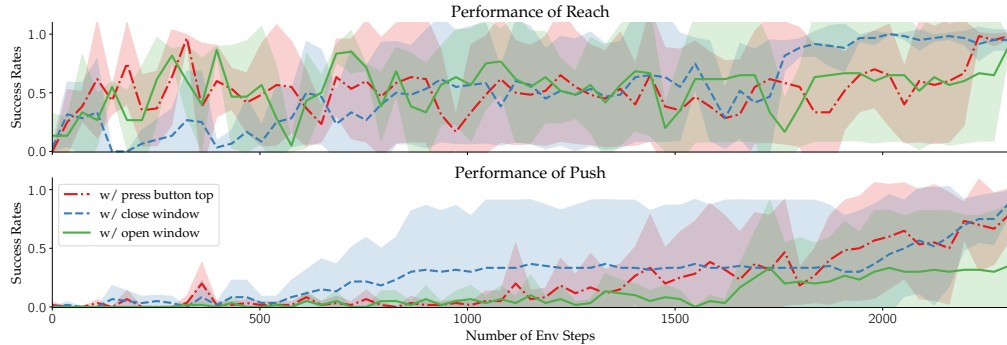

Figure 3: (Top) Performance of reach when co-trained with "push" and one of "press button top", "close window", or "open window". (Bottom) performance of "push" when co-trained with reach and one of "press button top", "close window", or "open window". In both cases, the best performance is achieved by co-training with "close window". Performance is degraded when co-trained with "open window".

denote the total loss where $L_i$ represents the non-negative loss of task $i$. For simplicity of notation, we set the loss weight of each task to be equal to 1, though our construction generalizes to arbitrary weightings.

For a given training batch $\mathcal{X}^t$ at time-step $t$, we can first update the task specific parameters $\{\theta_i^{t+1}\}$ using standard gradient updates. We can now define the quantity $\theta_{s|\xi}^{t+1}$ to represent the updated shared parameters after a gradient step with respect to the tasks in the non-empty subset $\varnothing \subset \xi \subseteq [m]$. Assuming SGD for simplicity, we have[1]

$$\theta_{s|\xi}^{t+1} := \theta_s^t - \eta \sum_{i \in \xi} \nabla_{\theta_s} L_i(\mathcal{X}^t, \theta_s^t, \theta_i^t).$$

This quantity allows us to calculate a *lookahead* loss using the updated shared parameters while keeping the task-specific parameters as well as the input batch unchanged across different subsets of task gradients. That is, in order to assess the effect of the gradient update of tasks in $\xi$ on a given task $j$, we can compare the loss of task $j$ before and after applying the gradient update on the shared parameters with respect to $\xi$. In order to eliminate the scale discrepancy among different task losses, we consider the ratio of a task's loss before and after the gradient step on the shared parameters as a scale invariant measure of relative progress. We can then define an asymmetric measure for calculating the *transference* of the tasks in $\xi$ at a given time-step $t$ on a single task $j$ as

$$\mathcal{Z}_{\xi \to j}^t = 1 - \frac{L_j(\mathcal{X}^t, \theta_{s|\xi}^{t+1}, \theta_j^{t+1})}{L_j(\mathcal{X}^t, \theta_s^t, \theta_j^t)}. \tag{1}$$

Notice that a positive value of $\mathcal{Z}_{\xi \to j}^t$ indicates that the update on the shared parameters results in a lower loss on task $j$ than the original parameter values, while a negative value of $\mathcal{Z}_{\xi \to j}^t$ indicates that the shared parameter update is antagonistic for this task's performance. Also, note that for $\xi = \{j\}$, our definition of transference encompasses a notion of *self-transference*, i.e. the effect of a task's gradient update on its own loss. This quantity is particularly useful as a baseline to determine whether a subset of gradient updates can result in improved performance when compared with a task's own self-transference. As we discuss in the next section, transference provides an effective guideline for choosing the subset of tasks to train together in a multi-task setting.

## 3.2 TASK GROUPINGS BASED ON TRANSFERENCE

Before using transference to develop a multi-task training augmentation, we aim to evaluate if our measure of transference is meaningful in practice. To do this, we empirically test whether transference is predictive of whether a group of tasks should be trained together. We consider two multi-task learning problems based on the CelebA dataset (Liu et al., 2015) and the Meta-World benchmark (Yu et al., 2019). Compiling transference scores into a radar chart, we use Figure 1 to identify groupings of tasks which exhibit beneficial or antagonistic transference. We then evaluate if our heuristic led us to select ideal task groupings by comparing against all other possible task groupings. Unlike prior approaches, our method requires only a single training run and is minimally complex, only making additional forward and backward passes through the network which can be done in parallel.

---

[1]Note that the backward pass is computed only once and the gradients are calculated at $\{\theta_s^t\} \cup \{\theta_i^t | i \in [m]\}$.

| Co-trained with | Group 1 | | | Group 2 | |
|---|---|---|---|---|---|
| | $A_0$ | $A_5$ | $A_4$ | $A_8$ | $A_7$ |
| {All Tasks} | 92.65 | 96.76 | 96.27 | 87.98 | 95.14 |
| $A_0$ | - | - | - | 88.00 | 94.92 |
| $A_1$ | 92.55 | 96.77 | 96.48 | 87.93 | **95.20** |
| $A_2$ | 92.56 | 96.77 | 96.51 | 88.04 | 95.02 |
| $A_3$ | **93.10** | **96.84** | **96.56** | 87.97 | 95.11 |
| $A_4$ | - | - | - | 87.92 | 95.10 |
| $A_5$ | - | - | - | 88.11 | 95.15 |
| $A_6$ | 92.63 | 96.54 | 96.14 | **88.46** | **95.20** |
| $A_7$ | 92.48 | 96.66 | **96.56** | - | - |
| $A_8$ | 92.71 | 96.69 | 96.34 | - | - |

Table 1: Test Accuracy on CelebA. Group 1 and Group 2 are co-trained with the task(s) in the left column. Group 1 most benefits from co-training with $A_3$ while Group 6 most benefits from co-training with $A_6$.

We first consider a multi-task classification problem by selecting 9 attributes[2] from the CelebA dataset and computing their transference when trained together in a single model. Specifically, we compute the transference of each task $i$ on all other tasks $j$ in the network, i.e. $\mathcal{Z}^t_{\{i\} \to j}$. While transference is computed at a per-minibatch level, we can average the transference across mini-batches to compute an epoch-level transference metric. Integrating across the number of steps in training then provides us with an overall (scalar) transference score. Figure 1(a) shows the transference score among the 9 attributes in the CelebA dataset. For purposes of illustration, we normalize the transference scores on each task by dividing the values by the task's self-transference. Thus, self-transference becomes 1 for all tasks.

As illustrated in Figure 1(a), two clusters of strong mutual transference manifest: (1) $\{A_0, A_3, A_4, A_5\}$ and (2) $\{A_6, A_7, A_8\}$. We draw this grouping by choosing subsets of tasks which induce relatively high mutual transference. For instance, $A_3$ demonstrates significantly higher transference on $A_0$, $A_4$, and $A_5$, when compared with the transference of $A_1$, $A_2$, $A_6$, $A_7$, and $A_8$ on these tasks. In Table 1, we construct Group 1 and Group 2 from interpreting Figure 1(a) and co-train both groups with all other attributes as shown in the left column of Table 1. We find the inclusion of $A_3$ in Group 1 ($A_0$, $A_4$, and $A_5$) results in the highest accuracy when compared to co-training with any other attribute. Similarly, $A_6$ is the best attribute to co-train with $A_7$ and $A_8$ in Group 2.

We also consider a multi-task reinforcement learning (RL) problem using the Meta-World benchmark (Yu et al., 2019), which contains 50 qualitatively different robotic manipulation tasks. We select five tasks from Meta-World task suite, namely "reach", "push", "press button top", "open window" and "close window". We train these five tasks together using the soft actor critic (SAC) (Haarnoja et al., 2018) algorithm with the weights of the critic and the policy shared across all tasks. We compute the transference on the critic loss to produce Figure 1(b). We include more details on the multi-task RL experiment in Appendix A.2.

Figure 1(b) indicates that "open window" exhibits relatively low transference with all tasks while "close window" exhibits especially high transference with "push" and "reach". Accordingly, we group "push" and "reach" together and then compute the efficacy of these tasks when co-training with "press button top", "open window", and "close window". As shown in Figure 3 and as suggested by transference, the success rate of "reach" converges more quickly to a significantly higher value when it is co-trained with "close window", and marginally faster when it is co-trained with "press button top", as compared to co-training with "open window". This effect is only magnified when we assess the performance of "push". For "push", its performance in terms of success rates and data efficiency is greatly increased when co-trained with either "close window" or "press button top" when compared to co-training with "open window".

In summary, transference can be used as a heuristic to determine task groupings. A set of tasks which exhibit relatively high transference tend to train effectively together, while tasks which exhibit relatively low transference with this set should be excluded. Using this method, our empirical analysis suggests transference is capable of identifying beneficial and antagonistic task groupings in both supervised and reinforcement learning paradigms.

---

[2]To avoid possible biases or implications conveyed by the definition of the attributes, we omit their names.

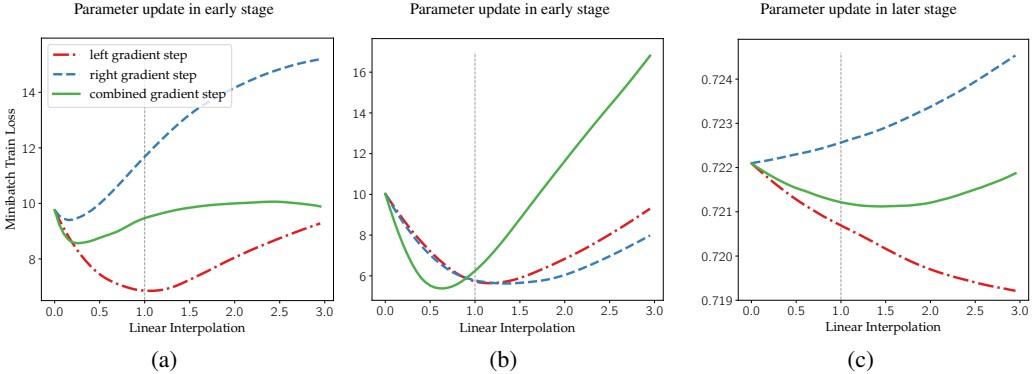

Figure 4: Total loss in MultiFashion along a linear interpolation between the current and updated shared parameter values. We extend the interpolation to a magnitude $3\times$ the original update to illustrate the curvature along this direction. The dashed vertical line crosses the loss curves at the original update. In all cases, a step along the left gradient is better.

## 4 INCREASED TRANSFER MTL

As shown in Section 3.2, the transference measure defined in Eq. (1) is an effective "macro-level" quantity to recognize the tasks that may benefit from co-training. In this section, we extend the utility of our transference measure beyond determining task groupings by incorporating it directly into the training dynamics of multi-task learning. In particular, we present a parameter-free algorithm which selects the combination of task gradients in each step of training that most increases transference among all tasks. Let us define *total transference* for the subset of tasks $\xi$ at time-step $t$ as

$$\mathcal{Z}_\xi^t := \sum_{j \in [m]} \mathcal{Z}_{\xi \to j}^t = \sum_{j \in [m]} \left( 1 - \frac{L_j(X, \theta_{s|\xi}^{t+1}, \theta_j^{t+1})}{L_j(X, \theta_s^t, \theta_j^t)} \right). \tag{2}$$

Total transference provides a cumulative measure of relative progress across all tasks as a result of applying a gradient update to the shared parameters with respect to a subset of tasks $\xi$. Perhaps surprisingly, $\xi = [m]$ is not always the set of tasks which most increases transference. Rather, we find that this particular update can often result in worse transference than a gradient update using a subset of tasks, an effect especially pervasive at the beginning of training.

With this motivation in mind, we present increased transfer multi-task learning (IT-MTL), a parameter-free augmentation to multi-task learning which chooses the gradient that most increases transference. Specifically, IT-MTL chooses the shared parameter update using a subset of tasks which induce the highest transference in a given minibatch. Formally, we define $\mathcal{J} \subseteq \mathcal{P}([m]) - \varnothing$ where $\mathcal{P}(\mathcal{S})$ denotes the power-set of set $\mathcal{S}$.[3] Although the possible number of task combinations is exponentially large in the number of tasks $m$, in practice and as found in our experiments in Section 5, a carefully chosen subset of tasks of size $|\mathcal{J}| = \mathcal{O}(m)$ provides compelling results. Specifically, choosing $\mathcal{J}$ as the set of $m$-many leave-one-out subsets, i.e. $[m] - \{i\}$ for all $i \in [m]$, plus the set of all tasks $[m]$ works well in practice. IT-MTL proceeds by calculating the total transference defined in Eq. (2) for each subset of tasks $\xi \in \mathcal{J}$ and then applies the gradient update to the shared parameters using the subset that induces the highest total transference. Task-specific parameters are updated as normal. The full algorithm is provided in Algorithm 1.

To further illuminate the intuition behind IT-MTL, we present a deeper analysis into the loss landscape of MultiFashion (Lin et al., 2019) in Figure 4. This figure provides insight into several cases where a single task gradient update on the shared parameters is more beneficial than the gradient using the full set of tasks. Figure 4(a) exemplifies the case where high curvature in the direction of the right task gradient significantly increases the total loss. Similarly, the combined gradient marginally decreases the total loss while the left gradient significantly decreases total loss. In a related instance, and as illustrated in Figure 4(b), high curvature in the combined gradient direction, but relatively low curvature in the direction of the left and right gradient, will also lead to a single task gradient exhibiting higher transference than the combined gradient.

---

[3]In other words, for a subset $\xi \in \mathcal{J}$ and $\xi \neq \varnothing$, either a particular task $i$ participates in the gradient update, i.e. $i \in \xi$, or not $i \notin \xi$.

---

**Algorithm 1** Increased Transfer Multi-Task Learning

---
1: Initialize network weights: $\{\theta_s\} \cup \{\theta_i \mid i \in [m]\}$
2: Set candidate subsets: $\mathcal{J} \subseteq \mathcal{P}([m]) - \varnothing$
3: **for** $t = 0, \ldots, T - 1$ **do**
4:      Compute per-task loss: $L_i(\mathcal{X}^t, \theta_s^t, \theta_i^t), \; \forall i \in [m]$      ▷ typical forward pass
5:      Update task-specific parameters: $\theta_i^{t+1} = \theta_i^t - \eta \nabla_{\theta_i} L_i, \; \forall i \in [m]$
6:      **for** $\xi \in \mathcal{J}$ **do**
7:          $\theta_{s|\xi}^{t+1} = \theta_s^t - \eta \sum_{i \in \xi} \nabla_{\theta_s} L_i(\mathcal{X}^t, \theta_s^t, \theta_i^t)$
8:          $\mathcal{Z}_\xi^t = \sum_{i \in [m]} \left( 1 - \frac{L_i(\mathcal{X}^t, \theta_{s|\xi}^{t+1}, \theta_i^{t+1})}{L_i(\mathcal{X}^t, \theta_s^t, \theta_i^t)} \right)$
9:      **end for**
10:     Select max transfer task combination: $\xi^\star = \arg\max_\xi \mathcal{Z}_\xi^t$
11:     Update shared parameters: $\theta_s^{t+1} = \theta_s^t - \eta \sum_{i \in \xi^\star} \nabla_{\theta_s} L_i(\mathcal{X}^t, \theta_s^t, \theta_i^t)$
12: **end for**

---

While the first two cases of high curvature occur predominantly during the early rounds of training, a third case which occurs throughout training is shown in Figure 4(c). In this instance, the right task's gradient marginally decreases its own loss but significantly increases the loss of the left task. This causes the combined gradient to only marginally decrease the total loss. On the other hand, the left gradient most increases transference. As a result, only using the left gradient significantly improves the total loss. Additional information regarding this analysis can be found in Appendix A.2.1.

## 4.1 A First Order Approximation of the Increased Transfer MTL Method

The IT-MTL procedure requires multiple forward-backward passes to calculate the lookahead losses of tasks in $\mathcal{J}$. This may become computationally prohibitive for large models, especially as the number of candidate tasks in $\mathcal{J}$ grows. In this section, we derive a simple first order approximation of IT-MTL which requires only a single forward-backward pass. Unlike Algorithm 1, the approximation does not update the task-specific parameters before computing the update to the shared parameters, effectively moving line 5 in Algorithm 1 to line 11. Ignoring the learning rate $\eta$ for simplicity, a first order Taylor series expansion of transference in Eq. (1) yields:

$$
\begin{aligned}
\mathcal{Z}_{\xi \to j}^t &= 1 - \frac{L_j(\mathcal{X}^t, \theta_{s|i}^{t+1}, \theta_j^{t+1})}{L_j(\mathcal{X}^t, \theta_s^t, \theta_j^t)} \\
&\approx 1 - \frac{L_j(\mathcal{X}^t, \theta_s^t, \theta_j^t) - \langle \nabla_{\theta_s} L_j(\mathcal{X}^t, \theta_s^t, \theta_j^t), \sum_{i \in \xi} \nabla_{\theta_s} L_i(\mathcal{X}^t, \theta_s^t, \theta_i^t) \rangle}{L_j(\mathcal{X}^t, \theta_s^t, \theta_j^t)} \\
&= \frac{\langle \nabla_{\theta_s} L_j(\mathcal{X}^t, \theta_s^t, \theta_j^t), \sum_{i \in \xi} \nabla_{\theta_s} L_i(\mathcal{X}^t, \theta_s^t, \theta_i^t) \rangle}{L_j(\mathcal{X}^t, \theta_s^t, \theta_j^t)},
\end{aligned}
$$

where $\langle \cdot, \cdot \rangle$ denotes inner product. Thus, total transference defined in Eq. (2) can be written as

$$
\mathcal{Z}_\xi^t = \sum_{j \in [m]} \mathcal{Z}_{\xi \to j}^t \approx \langle \sum_{j \in [m]} \frac{\nabla_{\theta_s} L_j(\mathcal{X}^t, \theta_s^t, \theta_j^t)}{L_j(\mathcal{X}^t, \theta_s^t, \theta_j^t)}, \sum_{i \in \xi} \nabla_{\theta_s} L_i(\mathcal{X}^t, \theta_s^t, \theta_i^t) \rangle,
$$

which can be rewritten as

$$
\begin{aligned}
&= \langle \nabla_{\theta_s} \sum_{j \in [m]} \log L_j(\mathcal{X}^t, \theta_s^t, \theta_j^t), \sum_{i \in \xi} \nabla_{\theta_s} L_i(\mathcal{X}^t, \theta_s^t, \theta_i^t) \rangle \\
&= \langle \nabla_{\theta_s} \underbrace{\log \prod_{j \in [m]} L_j(\mathcal{X}^t, \theta_s^t, \theta_j^t)}_{\text{log-product loss}}, \sum_{i \in \xi} \nabla_{\theta_s} L_i(\mathcal{X}^t, \theta_s^t, \theta_i^t) \rangle.
\end{aligned}
$$

Our IT-MTL approximation computes alignment between the gradients of the candidate tasks with the gradient of the first quantity in the inner product, which we call the "log-product" loss. The gradient of the subtasks with the strongest alignment to the gradient of the log-product loss is used to make the final update to the shared parameters. Note in the approximate procedure, the gradients are calculated once, and the approximation has computational complexity similar to that of gradient correction methods such as PCGrad (Yu et al., 2020) and GradNorm (Chen et al., 2017).

## 4.2 Affinity with Gradient Projection Methods

IT-MTL can be combined with related work which modifies gradient direction and/or magnitude. The modified gradient can be added to the set $\mathcal{J}$ in Algorithm 1 as an additional candidate gradient for the current minibatch. If the modified gradient increases the total transference more so than the gradient of the candidate tasks in $\mathcal{J}$, it is used to update the shared parameters. We explore this idea in our experiments by composing $\mathcal{J} = \{\text{total loss}, \text{PCGrad(total loss)}\}$ to select between the typical multitask gradient and the PCGrad gradient.

## 5 Experiments

Motivated by our analysis in Section 4, we study the utility of transference in selecting the combination of gradients which increases transference for each minibatch. Unlike our evaluation of transference in Section 3.2 on datstes with a large number of tasks, IT-MTL is most computationally efficient when the number of tasks is small. Accordingly, we focus our evaluation on datasets with either 2 or 3 tasks and perform our analysis on MultiMNIST, a multitask variant of the MNIST dataset (LeCun et al., 1998); MultiFashion, a multitask variant of the MNISTFashion dataset (Xiao et al., 2017); and NYUv2 (Silberman et al., 2012). Further, we found the rate at which a single-task gradient is chosen over the combined gradient to be higher during the beginning of training. With this observation as motivation, we evaluate how information transfer changes with respect to convergence.

## 5.1 Increased Transfer Multi-Task Learning Evaluation

To assess the efficacy of IT-MTL, we evaluate its performance on the MultiMNIST and MultiFashion datasets. To increase comparability, we run our experiments on the same datasets as in (Lin et al., 2019) with a multitask variant of LeNet (LeCun et al., 1998). Full experimental details are provided in Appendix A.2.1, and (Lin et al., 2019) can be referenced for dataset construction. The code used in generating experimental results is attached to the supplementary material part of our submission.

For both datasets, we compare against the corresponding single-task baselines, equal weight multi-task learning, PCGrad (Yu et al., 2020), MGDA-UB (Sener & Koltun, 2018), and Uncertainty Weighing (UW) (Kendall et al., 2018). Table 2 summarises our experimental results. Notably, we find IT-MTL improves both left and right image accuracy over the equal-weight MTL baseline on both MultiFashion and MultiMNIST by choosing the gradient combination which most increases transference. Aside from this augmentation, there is no difference between these two models. Moreover, the IT augmentation can be combined with prior approaches to dynamically reweigh the tasks or directly modify the gradient by choosing the gradient combination which most increases transference. In particular, our method, and its corresponding approximation described in Section 4.1, combined with uncertainty weights and PCGrad (IT-UW-PCGrad) achieves very strong performance on both datasets.

To further evaluate the robustness of IT-MTL, we assess its performance on the more challenging NYUv2 dataset with a Multi-Task Attention Network architecture (MTAN) (Liu et al., 2019). The

| Method | MultiFashion | | MultiMNIST | |
|---|---|---|---|---|
| | Left Image Acc | Right Image Acc | Left Image Acc | Right Image Acc |
| Single Task Models | $79.81 \pm 0.39$ | $78.64 \pm 0.62$ | $89.40 \pm 0.17$ | $87.80 \pm 0.22$ |
| MTL | $78.80 \pm 0.35$ | $77.92 \pm 0.32$ | $89.01 \pm 0.22$ | $86.11 \pm 0.15$ |
| PCGrad | $78.17 \pm 0.48$ | $77.20 \pm 0.56$ | $88.92 \pm 0.21$ | $86.61 \pm 0.22$ |
| IT-MTL | $79.30 \pm 0.30$ | $78.17 \pm 0.36$ | $89.12 \pm 0.21$ | $86.35 \pm 0.23$ |
| MGDA-UB | $79.25 \pm 0.69$ | $78.62 \pm 0.48$ | $89.38 \pm 0.16$ | $86.81 \pm 0.46$ |
| UW-MTL | $80.93 \pm 0.27$ | $80.11 \pm 0.15$ | $90.77 \pm 0.13$ | $88.36 \pm 0.12$ |
| UW-PCGrad | $80.81 \pm 0.18$ | $80.19 \pm 0.19$ | $90.77 \pm 0.11$ | $88.36 \pm 0.09$ |
| IT-UW-MTL | $80.94 \pm 0.22$ | $\mathbf{80.31 \pm 0.21}$ | $90.71 \pm 0.18$ | $88.51 \pm 0.13$ |
| IT-UW-PCGrad | $\mathbf{81.39 \pm 0.20}$ | $80.30 \pm 0.15$ | $\mathbf{90.94 \pm 0.11}$ | $\mathbf{88.61 \pm 0.14}$ |
| IT-UW-PCGrad[‡] | $80.99 \pm 0.17$ | $80.25 \pm 0.21$ | $\mathbf{90.92 \pm 0.09}$ | $\mathbf{88.60 \pm 0.14}$ |

Table 2: Test Accuracy on MultiMNIST and MultiFashion classification datasets. The test accuracy is averaged over 10 samples. We report standard error, and best results are highlighted in bold. The approximation described in Section 4.1 is denoted with [‡]. Our results indicate the IT augmentation can improve the performance of traditional MTL, uncertainty weights (Kendall et al., 2018), and PCGrad (Yu et al., 2020).

| Method | Segmentation (Higher Better) | | Depth (Lower Better) | | Surface Normal | | | | |
|---|---|---|---|---|---|---|---|---|---|
| | | | | | Angle Distance (Lower Better) | | Within $t°$ (Higher Better) | | |
| | mIoU | Pix Acc | Abs Err | Rel Err | Mean | Median | 11.25 | 22.5 | 30 |
| Split, Wide | 15.89 | 51.19 | 0.6494 | 0.2804 | 33.69 | 28.91 | 18.54 | 39.91 | 52.02 |
| Split, Deep | 13.03 | 41.47 | 0.7836 | 0.3326 | 38.28 | 36.55 | 9.50 | 27.11 | 39.63 |
| Dense | 16.06 | 52.73 | 0.6488 | 0.2871 | 33.58 | 28.01 | 20.07 | 41.50 | 53.35 |
| Cross-Stitch | 14.71 | 50.23 | 0.6481 | 0.2871 | 33.56 | 28.58 | 20.08 | 40.54 | 51.97 |
| MTAN | 20.91 | 56.45 | 0.6111 | 0.2592 | 31.36 | 27.14 | 18.71 | 41.94 | 55.04 |
| IT-MTL | 21.36 | 56.48 | 0.5921 | 0.2516 | 31.21 | 26.92 | 19.74 | 42.41 | 55.37 |
| PCGrad | 22.93 | 57.79 | 0.6224 | 0.2687 | 30.90 | 26.81 | 19.64 | 42.56 | 55.63 |
| IT-PCGrad | **23.55** | **58.48** | 0.5926 | 0.2539 | **30.40** | **26.05** | 21.11 | **43.96** | **56.79** |
| IT-PCGrad[‡] | 23.43 | 57.38 | **0.5909** | **0.24.98** | 30.47 | 26.08 | **21.44** | **43.96** | 56.63 |

Table 3: 13-class semantic segmentation, depth estimation, and surface normal prediction results on the NYUv2 validation dataset. Performance of (Split, Wide), (Split, Deep), Dense, and Cross-Stitch (Misra et al., 2016) as reported in (Liu et al., 2019). The symbol [‡] denotes the approximation described in Section 4.1

dataset is composed of RGB-D images of indoor scenes and supports modeling of 13-class semantic segmentation, true depth estimation, and surface normal prediction. We follow the procedure of (Liu et al., 2019) and directly utilize their framework to evaluate the performance of IT-MTL. For computational efficiency, we form $\mathcal{J} = \{$semantic + depth, semantic + normal, depth + normal, semantic + depth + normal$\}$ in IT-MTAN and $\mathcal{J} = \{$semantic + depth + normal, PCGrad gradient$\}$ in IT-PCGrad. Table 3 summarizes our experimental findings. We find IT-MTAN improves modeling performance across all measurements for segmentation, depth, and surface normal tasks as compared with the MTAN baseline. IT-PCGrad and the approximation IT-PCGrad[‡] demonstrate similar improvements when compared with the PCGrad-MTAN baseline. This result indicates the benefit of IT-MTL can hold for complex neural network architectures on a challenging real world dataset.

## 5.2 Effect of Convergence on Transference

In this section, we return our focus to CelebA to analyze the effects of model convergence on information transfer. As shown in Figure 2, transference is a highly dynamic process that is significantly affected by model convergence. In particular, we find the transference of $A_6$ on $A_8$ to be nearly identical to that of $A_8$'s self-transference during the first two epochs of training. This result indicates the information encapsulated in the gradient update of $A_6$ on the shared parameters is as effective at minimizing the loss of $A_8$ as its own gradient update. However, this effect is dampened throughout training with positive transference only manifesting at the beginning of training.

Interpreting our results in the context of CelebA, the model may learn the location of certain attributes in the beginning of training which are highly transferable to other related attributes. Once this fundamental structure is learned, gradients may encode increasingly task-specific information leading to lower positive information transfer among tasks. These observations lend weight to the development of flexible sharing architectures, in particular those which can quickly adapt to changing information transfer dynamics in the shared parameters throughout training.

## 6 Conclusion

In this work, we take a first step towards quantifying information transfer in multi-task learning. We develop a measure to quantify transference and leverage this quantity to determine which tasks should be trained together as well as develop a method which improves multi-task learning efficiency and performance.

Nonetheless, the method is not without its shortfalls. Using transference to select task groupings does not account for regularization-like effects inherent in multi-task learning. As a result, although a specific set of task groupings may exhibit high transference, there will be cases when this grouping is sub-par. Moreover, the transference radar charts are open to interpretation. While the charts provide flexibility in determining task groupings or identifying tasks which detract from co-training, they do not unequivocally produce a final ranking. With regards to IT-MTL, training time scales linearly with respect to the number of tasks if the *lookahead* loss computation is not run in parallel.

In spite of these detriments, we hope our analysis of information transfer in multi-task learning encourages further analysis into its training dynamics. Future work on transference can incorporate this measure into a continuous-space learning algorithm, or guide the development of flexible architectures to further improve multi-task learning performance.

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

# A APPENDIX

## A.1 EXPERIMENTAL DETAILS

In this section, we detail our experimental methodology with the goal of facilitating reproducibility. The code used to produce our experimental results can be found by accessing the Supplementary Material section of our OpenReview submission.

### A.1.1 CELEBA

Our experiments on CelebA are generated using a combination of Keras (Chollet et al., 2018) and TensorFlow (Abadi et al., 2016) and access the CelebA dataset publicly available on Tensor-Flow datasets https://www.tensorflow.org/datasets/catalog/celeb_a. We selected 9 attributes from the subset of 40 annotated attributes for our analysis.

The encoder architecture is based loosely on ResNet18 (He et al., 2016) with a shallow feed forward network decoder. A learning rate of 0.001 is used for 40 epochs with the learning rate halved at 30 epochs. The model uses a momentum optimizer with momentum set to 0.9 and a batch size of 256. We maintain a 5-epoch moving average of the task accuracies and report the highest average 5-epoch moving accuracy achieved during training.

We found our model exhibits similar, if not slightly improved, performance over the ResNet18 variant used in (Sener & Koltun, 2018) that was trained for 100 epochs; however given transference computes an update to the shared parameters, we adopted an architecture with less shared parameter capacity and more task-specific capacity to improve training time without sacrificing performance.

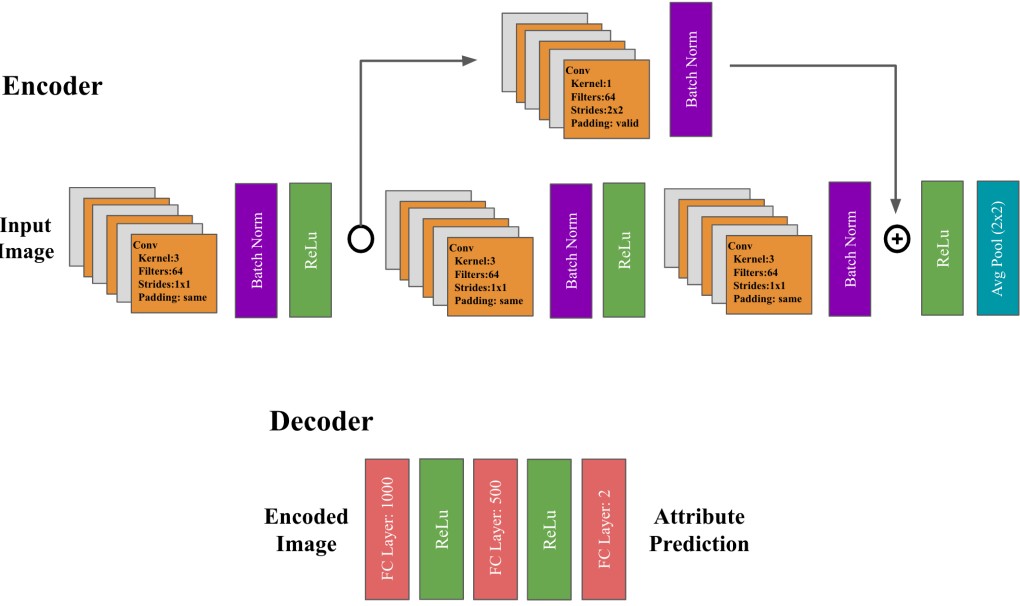

Figure 5: CelebA Encoder and Decoder used for measuring transference and determining which tasks should train together.

## A.2 META-WORLD

We use the five tasks: "reach", "push", "press button top", "open window", and "close window" from Meta-World (Yu et al., 2019). We use 6 fully-connected layers with 400 hidden units for both the policy and the critic with weights shared across all tasks. For each iteration, we collect 600 data points for each environment and train the policy and the critic for 600 steps with a batch size 128 per task. We use the soft actor critic (SAC) (Haarnoja et al., 2018) as our RL algorithm and adopt the default

Table 4: Chosen hyperparameters for MultiMNIST/Fashion experiments.

| Dataset | Method | lr | loss weight | GradNorm $\alpha$ |
|---------|--------|-----|-------------|-------------------|
| MultiFashion | MTL | $1e-3$ | 0.5 | — |
| | PCGrad | $1e-3$ | 0.5 | — |
| | IT-MTL | $1e-3$ | 0.5 | — |
| | UW-MTL | $1e-3$ | — | — |
| | IT-UW-MTL | $1e-3$ | — | — |
| MultiMNIST | MTL | $1e-3$ | 0.5 | — |
| | PCGrad | $1e-3$ | 0.5 | — |
| | IT-MTL | $1e-3$ | 0.5 | — |
| | UW-MTL | $1e-3$ | — | — |
| | IT-UW-MTL | $1e-3$ | — | — |

hyperparameters used in the public repository of SAC (`https://github.com/rail-berkeley/softlearning` at hash 59f9ad357b11b973b9c64dcfeb7cf7b856bbae91). We compute the transference on the critic loss

$$J_Q(\theta) = \mathbb{E}_{(\mathbf{s},\mathbf{a}) \sim \mathcal{D}} \left[ \frac{1}{2} (Q(\mathbf{s},\mathbf{a}) - \hat{Q}(\mathbf{s},\mathbf{a})^2) \right] ,$$

where $\mathbf{s}$ and $\mathbf{a}$ denote the state and action, $\hat{Q}$ denotes the target $Q$ network, $\mathcal{D}$ denotes the off-policy dataset collected by the agent, and $\theta$ denotes the parameter of the critic $Q$ network.

### A.2.1 MULTIMNIST/FASHION

Our experimental results on Multi-MNIST and Multi-Fashion are generated using a combination of Keras and TensorFlow. We evaluate on the datasets released by Lin et al. (2019) but further split $\frac{1}{6}$ of the training dataset into a validation set for final dataset splits of 100k/20k/20k train/valid/test.

The model architecture is loosely based on LeNet (LeCun et al., 1998) with a fully convolutional decoder and shallow feed-forward neural net decoder. A visual depiction is presented in (figure 6). The model uses a momentum optimizer with momentum set to 0.9 and a batch size of 256. The *lookahead* loss is computed by simulating the full momentum update to the shared parameters rather than the SGD update described in Section 3.1. The learning rate of the MTL baseline was selected on a validation dataset over $\{1e-4, 5e-4, 5e-3, 1e-2, 5e-2\}$ using a schedule which halves the learning rate every 30 epochs. A coarse grid search of the task-specific weights with left image weight = 1. - right image weight yielded left weight = right weight = 0.5. IT-MTL, Uncertainty Weight, and PCGrad used the same hyperparameters as the baseline. GradNorm was found to be much more sensitive to hyperparameters, and these were tuned via random search between $[1e-6, 1e-2]$ for the learning rate and $[1e-6, 5.0]$ for the spring constant. The parameters we used for each experiment are listed in Table 4.

Due to non-trivial inter-run variance, we ran each experiment to completion 6 times, dropped the worst performance, and averaged the remaining 5 runs to produce the results shown in Table 2. Moreover, we report the average accuracy of the final 5 epochs to further improve comparability.

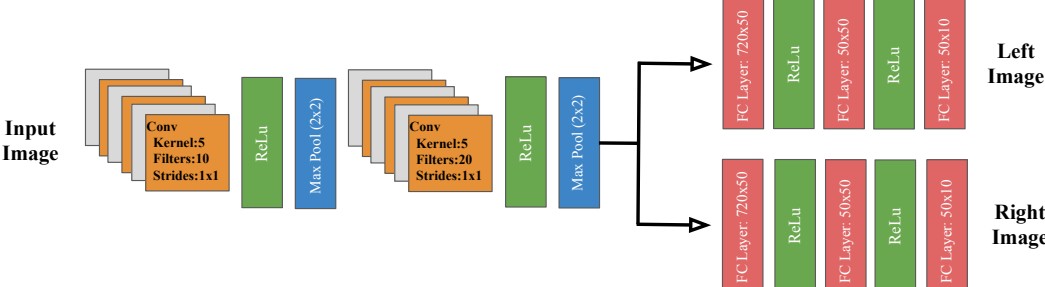

Figure 6: LeNet used in experiments. Note, the fully convolutional encoder and shallow neural net decoder.

## A.3 NYUv2

We clone the MTAN repository released by (Liu et al., 2019) (https://github.com/lorenmt/mtan at hash b6504093ea6d646dccf501bbbb525a4d85db96ba) to empirically test method and IT-PCGrad choosing between the combined gradient (i.e. gradient with respect to depth + semantic + normals loss) with the PCGrad gradient. The optimization uses Adam (Kingma & Ba, 2014), and the *lookahead* loss is computed by simulating the full Adam update to the shared parameters rather than the SGD update described in Section 3.1.

We run all MTAN experiments with default hyperparameters and settings with the exception of reducing the number of steps in PCGrad and IT-PCGrad to 100 as we find significant overfitting begins after this stage. Results from Split, Wide; Split, Deep; Dense; and Cross-Stitch results are taken from (Liu et al., 2019).

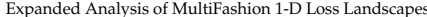

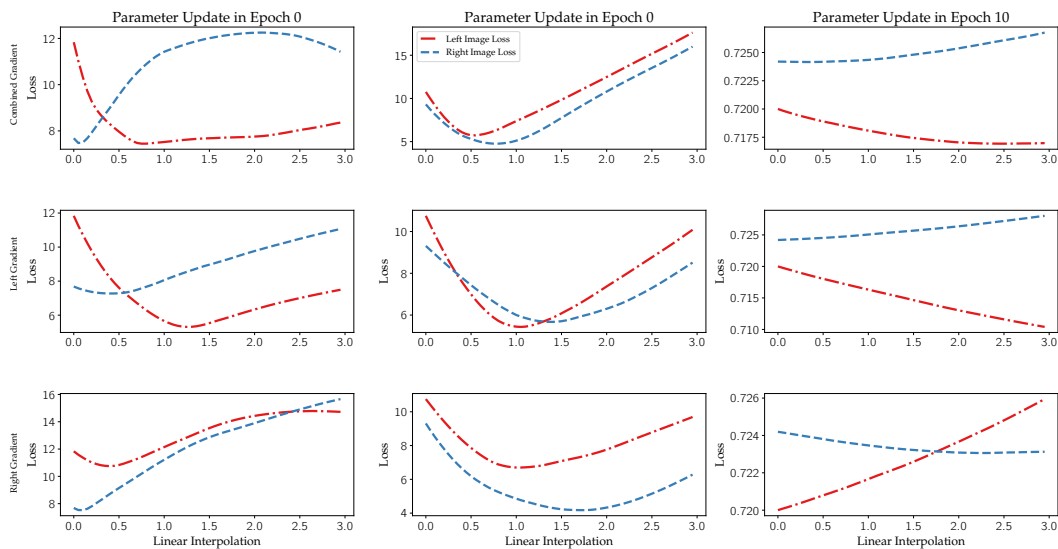

Figure 7: Expanded 1-d loss landscapes along each gradient direction.

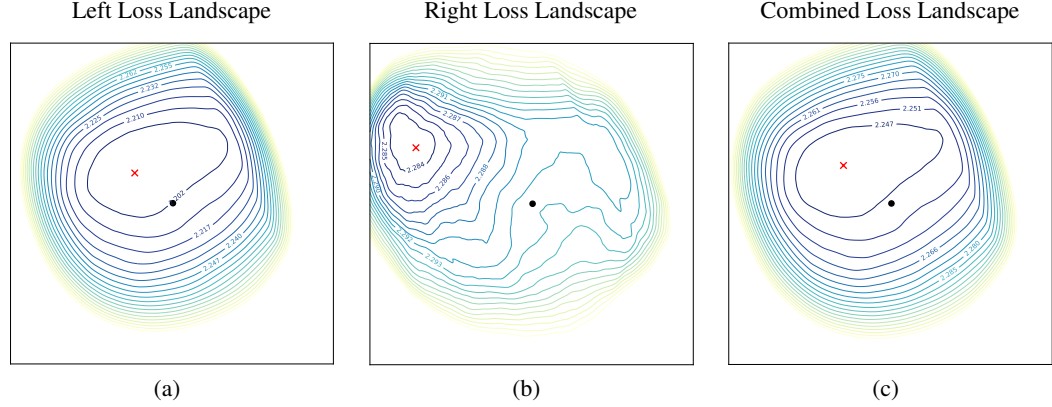

Figure 8: 2-Dimensional loss landscape of the (a) left digit loss, (b) right digit loss, and (c) the combined (average of left and right) loss in MultiFashion in a step where IT-MTL chooses the left gradient for the shared parameter update. The × symbol indicates the location of the minimum value of the loss in the local landscape and ● shows the projected coordinates of the current parameter.

## A.4 Loss Landscape Expanded Analysis

Figure 4 was created by halting training in a given epoch immediately after either the left or the right task gradient update manifests higher transference than the $1/2$(left + right) (i.e. combined) gradient update. We then applied the parameter update to the shared parameters using SGD with momentum to create a linear interpolation between the current parameter values and the parameter values following an update. We extend this interpolation 3x past the typical update to measure the curvature along the direction of the update.

In Figure 7, we compute the loss along a linear interpolation of the left gradient, the right gradient, and the combined gradient direction with each column corresponding to the total loss plot presented in Figure 4. For instance, Column 1, Row 2 plots the left and right loss along the left gradient step for the leftmost plot in Figure 4 and Column 2, Row 2 plots the left and right loss along the right gradient step for the center plot in Figure 4.

In Figure 8, we plot the 2-dimensional loss landscape of the left and right loss as well as the combined loss for MultiFashion. To generate the plots, we first sample two random directions in the parameter space and then scale the norms of these directions to be equal to the norm of the parameters. Next, we interpolate the parameters along these two directions in the range $[-0.1, +0.1]$ times the norm of the parameters.

The left image loss depicts a smooth landscape whereas the right image loss is highly non-smooth. Notice that the level sets of the combined (i.e. average) loss is higher than those of the left loss. For this step, IT-MTL chooses the left gradient for the shared parameter update which aligns with the curvature discrepancy between the right image loss and the left image loss.

