# OpenReview forum: "Measuring and Harnessing Transference in Multi-Task Learning"
_ICLR.cc/2021/Conference — Reject_

### Official Review · AnonReviewer4 · 2020-10-17
**The idea is interesting, but writing should be improved.**

**Rating:** 6
**Confidence:** 5

**Review:**

[Summary] This paper studies the problem of task relationship/transference in multi-task learning, by introducing a quantifiable measurement based on relative loss updates.  A (nonsymmetric) task transference between task $i$ and task $j$ then can be computed by measuring the relative change of training loss of task $j$, with the updated shared parameters from the training loss of task $i$. By finding a subset of tasks that achieves maximal total transference over every single task, multi-task learning performance can be further improved.

[Strength] Understanding multi-task relationships and transference is important to achieve a good multi-task learning performance and on multi-task architecture design. The proposed solution based on the relative loss update is intuitive, clean, and simple to implement.

[Weakness]
1. **Task selection.** To achieve a maximal performance improvement, it seems that we have to compute task transference from all possible combinations of task grouping, which is exponential based on the number of tasks. The authors claim that "a carefully chosen subset of tasks with $|\mathcal{J}| =  \mathcal{O}(m)$ provides reasonable improvements", but I cannot find any details on how these tasks are selected. Besides, in the experiment section, the number of tasks evaluated is no more than 3, and the improvements are quite marginal (mostly within 0.5%).

2. **Visualisations are unclear.** In Fig. 1, task transference is visualised by the pairing of all possible combination of two different tasks. But $\xi$ selects a subset of tasks, rather than just an individual task. So how can we know the transference for each individual task just by the transference for all tasks? Or in this case, $\xi$ is specifically designed to be a single task? The authors should elaborate on these details.

2. **Notations are unclear.** $\xi$ represents a non-empty subset of all task groupings. So why introduce another notation $\mathcal{J}$ in Algorithm 1 to include all possible task groupings, this looks quite redundant to me.


I hope the authors could elaborate on these questions and include more relevant details in the paper.

---

> ### Author Response · Authors · 2020-11-19
> **Response to Reviewer4**
>
> Thank you your review and enumerating these weaknesses. Your suggestions center reducing ambiguity and improving transparency in the text which we deeply appreciate. We have made improvements to the body of our work to elucidate the questions posed in your review in addition to answering them below.
>
> 1. **Task Selection.** While the choice of $\mathcal{J}$ can depend on learning paradigm and computational budget, we recommend the set of $m$-many leave-one-out subtasks where $m$ is the number of tasks in the network in addition to including the set $[m]$, i.e. all tasks, within $\mathcal{J}$ to represent the typical MTL update. For example, in MultiMNIST/Fashion, $\mathcal{J}$ can be constructed of \{left loss, right loss, left loss + right loss\}; while in NYUv2, we can increase computational efficiency constructing $\mathcal{J}$ of \{semantic + normal, semantic + depth, normal + depth, semantic + normal + depth\}. Alternatively if the PCGrad optimizer [1] is used, $\mathcal{J}$ = \{semantic + normal + depth, PCGrad(semantic + normal + depth)\} is sufficient to improve performance as shown in the IT-PCGrad rows of Table 3. We have modified Sections 4 and 5.1 to further expand upon this point. Moreover, we derive a first-order Taylor approximation of IT-MTL in Section 4.1 to improve computational efficiency which permits efficient scaling of IT-MTL to a large number of tasks.
> 2. **Visualizations.** As you posit, $\xi = \{i\}$ is designated as a single task when using transference to determine task groupings. We have made changes to Section 3.2 and the caption of Figure 1 to eliminate this ambiguity.
> 3. **Notations.** In Algorithm 1, $\mathcal{J}$ is the set of non-empty subsets of task groupings and $\xi$ is an element of $\mathcal{J}$ that includes a set of subtasks. For example, if a model makes predictions for Task 1, Task 2, and Task 3; we would use the $\binom{3}{2}$ subsets: $\xi_1$ = \{Task 1, Task 2\}, $\xi_2$ = \{Task 1, Task 3\}, $\xi_3$ = \{Task 2, Task 3\}. Then $\mathcal{J}$ would be $\{\xi_1, \xi_2, \xi_3\}$ + \{Task 1, Task 2, Task 3\}.
>
> Given the significant revisions to our paper -- both in terms of reducing ambiguity and improving the performance of IT-MTL by combining it with PCGrad -- we hope you will reconsider your initial rating within the context of these improvements. Please let us know if you have any other questions, and we will answer them and make appropriate changes to the paper's text.

---

> > ### Comment · AnonReviewer4 · 2020-11-24
> > **Response**
> >
> > Thanks for the response and further clarifications.
> >
> > The updated paper now looks significantly clearer.
> >
> > I have no further questions and thus maintain my score as weak accept.

---

> ### Comment · Area_Chair1 · 2020-11-23
> **Author response available**
>
> Dear reviewer,
>
> The authors have responded to your comments below. Could you please go over the response and give feedback to the authors sometime soon? The interactive discussion deadline is this Tuesday and you will not be able to interact with the authors after the date.
>
> Thanks,
> AC

---

### Official Review · AnonReviewer3 · 2020-10-28
**Interesting paper with major flaw**

**Rating:** 5
**Confidence:** 4

**Review:**

This paper studies the transferability in multi-task learning. They propose a metric, transference, to evaluate how tasks affect each other during multi-task training, and a method called IT-MTL which utilizes this metric to compute and improve lookahead loss changes. Although the proposed metric and method are interesting from a scientific point of view, there are a few key downsides (as the author themselves summarized in the conclusion) that require further investigation/improvements.

Pros:
1. The idea of using change in losses to capture inter-task transferability is new and could have potential impact in the community.

Cons:
1. The biggest concern is efficiency. The proposed method requires calculating lookahead loss for a set of combinations of tasks. While the author claim it only require O(m) possibilities empirically, it is not guaranteed for other datasets/settings. In fact, even with O(m) complexity could still be prohibitive for large models. To reveal the full picture, I recommend the author to demonstrate the actual time required for training for each method, and conduct an ablation analysis on the efficiency-performance trade-off.
2. From Table 2, I can only observe marginal improvements over prior methods. Considering the extra computational cost, it is hard to justify the effectiveness of the method itself.
3. The proposed method does not account for optimizers with ema gradients such as Adam, which may diminish the effect of selecting subset of tasks.

Other comments:
1. The idea of using lookahead method on shared layers in multi-task learning has been recently explored in [1]. Although they use a different optimization process, the high-level idea of improving transference/validation loss is shared. On the other hand, dropping gradients is also recently proposed [2]. Though concurrent, it still might be good to mention in the final version.
2. I would also recommend exploring some greedy strategies for the proposed method to be efficient. In other words, how to greedily reduce the search space of combination of tasks.
3. This is not very important, but what will happen if we directly use the transference metric to reweight tasks? For example, in multi-source transfer learning, can we utilize the metric proposed to decide what tasks to use?

[1] On Negative Interference in Multilingual Models: Findings and A Meta-Learning Treatment. Wang et al., EMNLP 2020.

[2] Just Pick a Sign: Optimizing Deep Multitask Models with Gradient Sign Dropout. Chen et al., NeurIPS 2020.

---

> ### Author Response · Authors · 2020-11-19
> **Response to Reviewer3**
>
> Thank you for your feedback -- it has been incredibly valuable in better understanding the primary message it conveys to a reader with an unbiased viewpoint.
>
> Simply put, you're right -- IT-MTL is very computationally expensive. Our initial goal with IT-MTL was to juxtapose the improvements from simply removing a task's gradient for certain minibatches with our surprise at this effect, rather than develop computationally efficient method. We also provided an empirical analysis into loss landscapes to explain this effect, which we believe would be useful to the MTL community. Thanks to your concerns over efficiency, we have derived a first-order Taylor approximation with computational complexity similar to that of PCGrad [3] and GradNorm [4]. The empirical performance of the approximation is included in Table 2 and Table 3 and is similar to that of the non-approximated method. We also explore combining IT-MTL with recent work (notably [3]) to further improve performance and find IT-MTL's affinity with related methods results in very strong experimental performance.
>
> Addressing your last con (\#3), in our empirical analysis we did account for optimizers with ema gradients. That said, we did not make this clear in our initial draft and have added a note in experimental details section of the appendix to elucidate this point. While the approximation presented in Section 4.1 does not take into account non-SGD optimizer update rules, we empirically find the performance of the approximation does not significantly degrade.
>
> With regards to your other comments,
> 1. We have included [1] and [2] in our related work section. As you note, the optimization is different ([1] differentiates through the first-order gradient update in a meta-learning algorithm similar to MAML to learn language-specific ``knowledge'' parameters). With regards to GradDrop, it will be interesting to see how their modified gradient compares to that of PCGrad [3] (or the typical combined gradient) as measured by Transference.
> 2. The main computational inefficiency lies in performing an update to the shared layers, and then recalculating the loss by making a second forward pass through the network. We have derived a Taylor approximation to avoid an update to the shared layers and subsequent second forward pass for each element of $\mathcal{J}$. The approximation has computational complexity similar to that of PCGrad and GradNorm [4].
> 3. We have been actively exploring this topic in a follow-up research effort, especially the concept of which task(s) to use at what stage of convergence (as suggested by transference).
>
> We hope you will reconsider your initial rating in the context of our revised paper, and please let us know if you have additional follow-up questions or concerns.
>
> [1] On Negative Interference in Multilingual Models: Findings and A Meta-Learning Treatment. Wang et al., EMNLP 2020.
>
> [2] Just Pick a Sign: Optimizing Deep Multitask Models with Gradient Sign Dropout. Chen et al., NeurIPS 2020.
>
> [3] Gradient Surgery for Multi-Task Learning, Yu et al., NeurIPS, 2020.
>
> [4] Gradient Normalization for Adaptive Loss Balancing in Deep Multitask Networks, Chen et al., ICML 2018.

---

> > ### Comment · AnonReviewer3 · 2020-11-23
> > **Follow up**
> >
> > Thank you for taking your time to address my concerns. I appreciate your effort for deriving an approximation and carrying it out during the rebuttal period. Your revision does address some of my concerns so I will raise my score. That said, as mentioned by other reviewers, the updated results are still not strong enough and the actual training time not clear. So I would still lean towards rejection. However, I do recognize your effort and the potential of this paper, but I just feel that it can still largely benefit from another round of major revision before final publication.

---

> > > ### Author Response · Authors · 2020-11-23
> > > **Clarification on Training Time and Experimental Analysis**
> > >
> > > We have evaluated transference on 5 separate multi-task datasets.
> > >
> > > In CelebA and Meta-World, we find transference can be used as a heuristic to find ideal task groupings (Table 1 and Figure 3 respectively). In Multi-MNIST, Multi-MNISTFashion, and NYUv2, we find choosing the gradient which increases transference greatly improves results (Table 2 and Table 3 respectively). We refer readers to Figure 4 of [1] for past experimental results on Multi-MNIST and Multi-FashionMNIST (we use the exact same dataset) to show how small the differences are among methods on both these datasets. We also refer the reviewers to Table 3 of [2] for past NYUv2 results (again, we use the exact same dataset).
> > >
> > > With regards to ambiguity in the actual training time, the approximation we derived makes a single forward and backward pass through the network for each minibatch. It has complexity similar to that of PCGrad as both methods compute the dot product among a set of |n| task gradients, and then chooses a final gradient based on this comparison.
> > >
> > >
> > > $m$ = number of tasks in the network
> > >
> > > $|\mathcal{J}|$ = number of candidate tasks $\mathcal{O}(m)$
> > >
> > > | Method / Computation | Baseline Multitask | &nbsp;&nbsp;&nbsp;&nbsp;&nbsp;&nbsp;&nbsp;PCGrad | IT-MTL  | IT-MTL (approx)|
> > > |:-  | :-: | :-: | :-: | :-: |
> > > | Num. forward props | 1 | 1 | $m$ | 1 |
> > > | Num. backward props | 1 | 1 | $m$ | 1 |
> > > | Auxiliary computation | 0 | $\frac{m^2}{2}$ dot products | 0 | $m$ dot products |

---

> ### Comment · Area_Chair1 · 2020-11-23
> **Author response available**
>
> Dear reviewer,
>
> The authors have responded to your comments below. Could you please go over the response and give feedback to the authors sometime soon? The interactive discussion deadline is this Tuesday and you will not be able to interact with the authors after the date.
>
> Thanks,
> AC

---

### Official Review · AnonReviewer2 · 2020-10-29
**Paper is well written. More comparison is needed. The significance of the proposed method is questionable.**

**Rating:** 4
**Confidence:** 4

**Review:**

In this paper, the authors derived a quantitative measure of transference among tasks in the setting of multi-task learning. Based on this measure, two methods were subsequently proposed, one to find group of tasks that may benefit from collaborative training, the other to combine task gradients at each training step.

Despite there are typos in a few places, the paper is good written and easy to follow.

No comparison with any baselines was done to show the advantage of the proposed method to identify task groups. One such baseline can be first training each task separately, then for each trained network fixing the shared parameters, resetting and adapting the task specific parameters for other tasks, and cross comparing the performance of all obtained models to identify task group for which collaborative training could be beneficial for individual tasks in the group.

I was not able to find any guidelines in the paper for how to construct the candidate set J. So, it would be most likely exhaustive search? I guess because of this, as the authors mentioned, their method is not scalable, only can handle problems containing small number of tasks. The empirical results are very weak. With considering the variance, I doubt any of the improvement in Table 2 is statistically significant.  Such weak results do not justify the significantly elevated computational cost. Based on the results included in the paper, I do not think the proposed method has any practical significance.

---

> ### Author Response · Authors · 2020-11-19
> **Response to Reviewer2**
>
> We sincerely appreciate your feedback and have made improvements to the paper to address the points you raise.
>
> With regards to identifying task groupings, the baseline you mention in your review would require $\mathcal{O}(m^2)$ consecutive runs (a total of 72 runs on CelebA and 20 on Meta-World) to build the radar charts presented in Figure 1. On the other hand, Transference requires a single training run to determine task groupings. We hope our comparisons in Figure 3 and Table 1 (i.e. exhaustively comparing against a held-out task for a given grouping) is sufficient to validate the effectiveness of using Transference to select task groupings.
>
> In response to your second comment, it depends on the learning paradigm and computational budget. For MultiMNIST/Fashion, $\mathcal{J}$ can be constructed of \{left loss, right loss, left loss + right loss\}; while in NYUv2, we can increase computational efficiency constructing $\mathcal{J}$ of \{semantic + normal, semantic + depth, normal + depth, semantic + normal + depth \}. In short, we typically use $\vert\mathcal{J}\vert = \mathcal{O}(m)$, i.e. linear in number of total tasks, in our experiments. Alternatively if the PCGrad update [1] is used, $\mathcal{J}$ = \{semantic + normal + depth, PCGrad(semantic + normal + depth)\} is sufficient to improve performance as shown in the IT-PCGrad and its approximation rows of Table 3. We have added Section 4.2 and have rewritten portions of Sections 4 and 5.1 to make this point more clear.
>
> Thanks to your review pushing us to improve IT-MTL: we've derived a first order Taylor approximation to decrease the computational burden of IT-MTL. We also include analysis into the affinity of IT-MTL with PCGrad and Uncertainty Weights [1] and empirically find IT-MTL has a beneficial effect on PCGrad + Uncertainty Weights. Additionally, our approximation allows scaling to a significantly larger set of candidates $\mathcal{J}$. Please see our meta comment for additional context, and let us know if you have any additional questions or concerns.
>
> [1]. Multi-Task Learning Using Uncertainty to Weigh Losses for Scene Geometry and Semantics, Kendall et al., CVPR 2018.
>
> [2]. Gradient Surgery for Multi-Task Learning, Yu et al., NeurIPS, 2020.

---

> > ### Comment · AnonReviewer2 · 2020-11-23
> > **Comments on author response**
> >
> > Even though only one single model needs to be trained to identify optimal task grouping in the proposed method, there are excessive forward and backward propagations for computing the transference metric. Thus, the proposed method may not be necessarily significantly more efficient than the baseline that I brought up in my initial review. I still think a comparison should be done to demonstrate the advantage of the proposed method, not just in terms of efficiency but also the accuracy in identifying the optimal task grouping.  In addition, the empirical results of the paper remain limited. It is difficult to tell for Table 3, since no standard deviation is provided. But in Table 2, most of the improvement does not seem to have statistical significance.

---

> > > ### Author Response · Authors · 2020-11-23
> > > **Computational Efficiency of Determining Task Groupings**
> > >
> > > **On the computational efficiency for the naive baseline vs transference to determine task groupings:**
> > >
> > > For Celeb-A the runtimes are as follows:
> > > the naive baseline for computing task grouping takes 41,130 seconds whereas transference takes 8,330 seconds.
> > >
> > > The baseline requires the practitioner to first train a single task model, freeze the typical shared layers, reinitialize the typical task specific layers, and then retrain the task-specific layers for each other task in the network. This complete process is then repeated for every other task in the network. On the other hand, transference trains all tasks together in a single run.
> > >
> > > Then the baseline has complexity (num tasks) * [(time to train a single task) + [time for fine tuning * (num tasks - 1)] whereas the complexity of transference is (time to train a multi-task model) +  (time to do an additional forward/backward pass) * (num task).
> > >
> > > On a Tesla V100 GPU for CelebA, it takes approximately 10 seconds to load the dataset and 208 seconds for a single epoch of training all tasks together while computing transference. We run our model for 40 epochs resulting in a final time of 8,330 seconds.
> > >
> > > On the other hand, it takes 44 seconds to complete a single epoch of a single-task model without computing transference statistics. When you don’t need to backprop through the ResNet layers (i.e. during the fine-tuning phase on another task), this time drops to 36 seconds to complete a single epoch. We make the assumption it will take 10 epochs of training to fine tune the task specific parameters from reinitialization.
> > >
> > > Then it will take approximately  44 * 40 + 8 * 10 * 36 + 10 = 4,570 seconds to evaluate the directional “similarity” of one task on all other tasks in our dataset. With 9 tasks in our dataset, it will take 41,130 seconds which is approximately 5 times as long as using transference. This difference is only exaggerated as the number of tasks increases as the baseline grows quadratically with respect to the number of tasks while transference grows linearly.
> > >
> > > **On the accuracy of identifying ideal task groupings:**
> > >
> > > On Meta-World, we choose to train “reach” and “push” together and then exhaustively evaluate the performance of all other tasks with this task grouping. Given our exhaustive analysis, we find transference selects the ideal task grouping: first close window, second push button top, and third open window. We perform a similar analysis on CelebA with regards to selecting tasks for groups which exhibit high self-transference, and again via an exhaustive search find transference selects the best task for these groups.
> > >
> > > We compare using transference to select task groupings against an exhaustive search. Evaluating another method of selecting task groupings can not perform better than an exhaustive search (which is the ground truth).
> > >
> > > **On the significance of the Table 2 and Table 3 results:**
> > >
> > > In Multi-MNIST, Multi-MNISTFashion, and NYUv2, we find choosing the gradient which increases transference greatly improves results (Table 2 and Table 3 respectively). We refer readers to Figure 4 of [5] for past experimental results on Multi-MNIST and Multi-FashionMNIST (we use the exact same dataset) to show how small the differences are among the methods we evaluate. We also refer the reviewers to Table 3 of [3] for past NYUv2 results (again, we use the exact same dataset).
> > >
> > > Our results on MultiMNIST and Multi-MNISTFashion are averaged across 10 subsequent independent runs. A clear trend emerges from the results which shows a clear and consistent ordering of the method’s performance -- notably IT-UW-PCGrad-MTL > IT-UW-MTL > UW-MTL > MGDA > IT-MTL > MTL -- on both datasets.
> > >
> > > Related to the standard deviation of Table 3, similar to [1], [2], [3], and [4], we do not compute standard deviation measurements for our results on NYUv2. We found the computational burden of running this dataset multiple times to be significant, especially when the difference among MTAN, IT-MTAN, PCGrad, and IT-PCGrad is apparent.
> > >
> > > [1] AdaShare: Learning What to Share For Efficient Deep Multi-Task Learning, Sun et al., NeurIPS 2020
> > >
> > > [2] Gradient Surgery for Multi-Task Learning, Yu et al., NeurIPS, 2020.
> > >
> > > [3] End-to-End Multi-Task Learning with Attention, Liu et al., CVPR 2019
> > >
> > > [4] Dense 3D semantic mapping of indoor scenes from RGB-D images, Hermans, et al., IEEE 2014
> > >
> > > [5] Pareto Multi-Task Learning, Lin et al., NeurIPS 2019

---

### Official Review · AnonReviewer1 · 2020-10-29
**The paper is not technically sound.**

**Rating:** 4
**Confidence:** 5

**Review:**

This paper proposes some interesting observations and proposes a novel measure of transference. Based, the proposed measure, this paper proposes a task-grouping method and novel training algorithm for MTL. However, there are some flaws in the proposed method and the proposed methods seems not technically sound. My concerns are listed as follows.

1. The definition of the transference measure is problematic. The measure is depended on the empirical loss. However, in MTL, the negative transfer or positive transfer is proposed with respect to the generalization loss. The increasing empirical loss may not lead to an increasing generalization loss. This paper should give theoretical support to show that the proposed measure can indicate the influence on generalization loss.

2. This paper does not provide experimental sufficient support for the superiority of the proposed method. For example, this paper is closely related to the previous work [1]. However, this paper has not compared with it. The authors should add the comparison with [1].

3. The paper is hard to follow. Some important details are not clear. For example, how to do task grouping based on the measure of transference are not clearly written.

[1]. Yu T, Kumar S, Gupta A, et al. Gradient surgery for multi-task learning. NeurIPS, 2020.

---

> ### Author Response · Authors · 2020-11-19
> **Response to Reviewer1**
>
> Thank you for raising these concerns. We address each of them below.
>
> 1.  While you are correct stating multi-task learning can improve generalization [0], prior works such as [1], [2], [3], [4], and [5] address negative transfer within the context of reducing training loss and improving learning efficiency rather than focusing on generalization performance. Similar to our work, they focus on the training-time gradients to improve optimization dynamics which leads to improved learning efficiency and final model performance.
> 2. We did compare with Gradient Surgery for Multi-Task Learning in Table 2 (row PCGrad and UW-PCGrad) in the original submission. In the revision, we've added Section 4.2 to expand on combining IT-MTL with methods such as [2] and have also added a comparison with [2] in Table 3 as well.
> 3. We suggested using transference as a heuristic to select task groupings such that tasks which exhibit high transference are grouped together (see Section 3.2: Task Groupings Based on Transference). We have revised Section 3.2 to make this more clear and have revised Section 4 to improve the overall readability of the paper.
>
> In light of our paper revisions, please let us know if you have any additional comments or concerns.
>
> [0]. On Negative Interference in Multilingual Models: Findings and A Meta-Learning Treatment. Wang et al., EMNLP 2020.
>
> [1]. A Modulation Module for Multi-Task Learning with Applications in Image Retrieval, Zhao et al., ECCV 2018.
>
> [2]. Gradient Surgery for Multi-Task Learning, Yu et al., NeurIPS, 2020.
>
> [3]. Gradient Normalization for Adaptive Loss Balancing in Deep Multitask Networks, Chen et al., ICML 2018.
>
> [4]. Gradient Vaccine: Investigating and Improving Multi-task Optimization in Massively Multilingual Models, Wang, et al., arXiv:2010.05874.
>
> [5]. On Negative Interference in Multilingual Models: Findings and A Meta-Learning Treatment. Wang et al., EMNLP 2020.

---

> ### Comment · Area_Chair1 · 2020-11-23
> **Author response available**
>
> Dear reviewer,
>
> The authors have responded to your comments below. Could you please go over the response and give feedback to the authors sometime soon? The interactive discussion deadline is this Tuesday and you will not be able to interact with the authors after the date.
>
> Thanks,
> AC

---

### Author Response · Authors · 2020-11-19
**Meta Comment To All Reviewers**

We would like to thank the reviewers for their time and suggestions to improve our work. Our primary goal in this paper was to formulate a measure that encapsulates the changing inter-task dynamics in MTL, empirically validate its effectiveness, and share our insights with the MTL community. Given the initial scores of [4, 4, 4, 6] and reviewer feedback, we now recognize the computational inefficiencies and incremental improvement of IT-MTL were the two main weaknesses of the original submission.

We have revised the paper to address these concerns. In particular, we include additional experimental results in Section 5.1 combining IT-MTL with Uncertainty Weights [1] and PCGrad [2]. This combined method exhibits very strong empirical results as shown in Table 2 and Table 3. By modifying LeNet-5 in the MultiMNIST experiments (moving the 720 task-specific layer to the shared base) and averaging across 10 experimental runs rather than 5, we improve the stability and decrease the inter-run variability of Table 2 results.

In Section 4.1, we derive a first-order Taylor approximation of IT-MTL that has computational efficiency similar to that of PCGrad [2] and GradNorm [3]. We report the performance of this approximation in Table 2 and Table 3, and find it to be similar to that of the non-approximated method.

Finally, we have added Sections 4.1 and 4.2 in addition to rewriting parts of Sections 2, 3.2, 4, A.2.1, and A.3 (changes in red) to address the ambiguities in notations and visualizations highlighted by the reviewers.

We would like to again thank the reviewers for pushing us to improve our method (both in terms of learning efficiency and computational complexity) and ask them to reconsider their initial ratings with regards to the revised paper.

[1] Multi-Task Learning Using Uncertainty to Weigh Losses for Scene Geometry and Semantics, Kendall et al., CVPR 2018.
[2] Gradient Surgery for Multi-Task Learning, Yu et al., NeurIPS, 2020.
[3] Gradient Normalization for Adaptive Loss Balancing in Deep Multitask Networks, Chen et al., ICML 2018.

---

> ### Author Response · Authors · 2020-11-23
> **Follow-up Meta Comment to All Reviewers**
>
> The primary concerns raised by Reviewer2 and Reviewer3 relate to the computational cost when computing task groupings in addition to using transference to improve multi-task learning efficiency (IT-MTL). As noted in our paper, as well as our reply to the reviewers, our approximation (Section 4.1) allows for significantly reducing computation and has complexity similar to that of PCGrad. Moreover, using transference to determine task groupings is significantly more efficient than the naive baseline proposed by Reviewer2.
>
> We sincerely thank the reviewers for their comments -- they have pushed us to derive an efficient method for IT-MTL as well as improve the overall readability of our paper. In light of the improved computational efficiency of our approximation and our most recents comments, we would appreciate additional feedback if time allows.

---

### Comment · Area_Chair1 · 2020-11-23
**The end of the discussion phase approaching**

Dear Reviewers,

The authors have provided detailed responses to your comments. Could you please go over the responses from the reviewers and provide feedback since the authors can have interactions with you only by this Tuesday (24th)?. I sincerely thank you for your service in reviewing for ICLR.

Thanks, Area Chair

---

### Decision · Program_Chairs · 2021-01-07
**Final Decision**

**Decision:**

Reject

**Comment:**

This paper proposes a method to quantify transference, which is a measure of information transfer across tasks, for multi-task learning framework. Specifically, the transference is measured as the change in the loss for a specific task after performing a gradient update for another. The proposed transference measure is used to both understand the optimization dynamics of MTL and improve the MTL performance, either by grouping tasks or combining task gradients based on the transference. The method is validated on multiple datasets and is shown to bring in some performance gains over the base MTL model (PCGrad, UW-MTL).

The majority of the reviewers were negative about this paper (4, 4, 5), while one reviewer gave it a positive rating (6). The reviewers in general agreed that the idea of measuring transference as the change in the loss with gradient updates is novel and intuitive. Yet, the reviewers had common concerns on the 1) weak performance improvements, and the 2) high-cost of computing the transference. While computing the transference requires additional computations with linear time complexity, which may be problematic with a large number of tasks, the performance gains using it were rather marginal (less than 0.5% over the baselines).  Another common concern from the reviewers was its insufficient experimental validation, as a comparative study against existing works that perform task grouping is missing. Both the authors and reviewers actively participated in the interactive discussion. However, the reviewers found that the two critical limitations persist even after the authors’ feedback, and in a subsequent internal discussion, they reached a consensus that the paper is not yet ready for publication.

Thus, although the proposed method is novel and appears to be promising, it may need more developments to make it both more effective and efficient. Moreover, there should be more in-depth analysis of its time-efficiency, and other benefits (e.g. interpretability) that could be achieved with the proposed transference measure. Finally, while there exist many works on learning both hard or soft task grouping, the authors do not reference or compare against them. To name a few, [Kang et al. 11] propose how to learn the discrete task groupings, [Kumar and Daume III 12] propose to learn a soft grouping between tasks, [Lee et al. 16] propose to learn soft grouping based on asymmetric knowledge transfer direction across the tasks, and [Lee et al. 18] proposes the extension of [Lee et al. 16] to a deep learning framework. I suggest the authors to discuss and compare against the above mentioned works, and fortify the related work section by searching for more classical works on multi-task learning.

- [Kang et al. 11] Learning with Whom to Share in Multi-task Feature Learning, ICML 2011
- [Kumar and Daume III 12] Learning Task Grouping and Overlap in Multi-task Learning, ICML 2012
- [Lee et al. 16] Asymmetric Multi-task Learning based on Task Relatedness and Confidence, ICML 2016
- [Lee et al. 18] Deep Asymmetric Multi-task Feature Learning, ICML 2018.